# Simple Application and Adherence to Gout Guidelines Enables Disease Control: An Observational Study in French Referral Centres

**DOI:** 10.3390/jcm11195742

**Published:** 2022-09-28

**Authors:** Charlotte Jauffret, Sébastien Ottaviani, Augustin Latourte, Hang-Korng Ea, Sahara Graf, Frédéric Lioté, Thomas Bardin, Pascal Richette, Tristan Pascart

**Affiliations:** 1Rheumatology Department, Saint-Philibert (GHICL), Lille Catholic University, 59160 Lille, France; 2Rheumatology Department, CHU Bichat (APHP), University Paris, 75018 Paris, France; 3Rheumatology Department, UMR 1132-Bioscar (Centre Viggo Petersen), CHU Lariboisière (APHP), University Paris, 75010 Paris, France; 4Biostatistics Department, Delegation for Clinical Research and Innovation, Lille Catholic University, 59160 Lille, France; 5ULR 4490-MABLab, CHU Lille, Lille University, 59000 Lille, France

**Keywords:** gout, allopurinol, febuxostat, urate-lowering therapy, anakinra, guidelines, treat-to-target, clinical inertia

## Abstract

Background: In a context of therapeutic inertia, the French Society of Rheumatology (SFR) published its first recommendations on gout in 2020, which were deliberately simple and concise. The objectives of the study were to determine the profile of patients referred to French gout-expert centres, and to examine the results of their management and the factors leading to those results. Methods: Three hundred patients attending a first visit for gout management in three French referral centres were retrospectively and randomly included in this multicentre observational study. Visits were performed at baseline (M0) and scheduled for month 6 (M6), month 12 (M12), and month 24 (M24). Results: Patients were 81% male and had a mean age 62.2 ± 15.2 years. Management followed French recommendations after the baseline visit in 94.9% of cases. SU levels were below 6.0 mg/dL in 59.4% of patients at M6, 67.9% at M12, and 78.6% at M24, with increasing clinical improvement (i.e., flare decrease) over 2 years of follow-up. At M24, 50% of patients were treated with allopurinol (313 ± 105 mg/d), which exceeded renal restrictions of doses in 61.5% of them, and 48.2% received febuxostat (84 ± 36 mg/d). The need for a sufficient dosage of ULT was the only predictive factor found for successful achievement of SU levels < 6.0 mg/dL at a given visit. Conclusions: Simple application of gout-management guidelines is feasible in clinical practice and is efficient, with a majority of patients achieving SU targets and clinical improvement.

## 1. Introduction

Gout is the only crystal-related arthropathy with available treatments to obtain crystal dissolution, and is the most common inflammatory arthritis in men over 40 years of age [1,2,3]. Currently, its prevalence is increasing worldwide [1], and it represents a major public-health and economic burden [2,4,5,6]. Its management is straightforward in the vast majority of patients. Urate-lowering therapies (ULT), mainly allopurinol and febuxostat, and flare/prophylaxis treatments such as colchicine, are simple and inexpensive [7]. When correctly prescribed and taken, these drugs enabled remission in almost all patients included in clinical trials, i.e., serum-urate (SU) level at target and no flares [8]. Nevertheless, due to a variety of patient-dependent and physician-dependent factors, including therapeutic inertia, gout remains poorly managed in primary care [6]. Real-life observational data show that the majority of patients do not reach targeted SU levels and still experience recurrent flares [6,9,10]. Under-prescription of ULT, poor treatment compliance, and fears and beliefs often insufficiently addressed in patient education have been identified as the main barriers for optimal care [11].

Recommendations from several international societies have recently been (re-)edited to provide guidance for the management of gout [3,12,13,14]. The 2020 French guidelines [3,13], very similar to the 2016 EULAR recommendations [12], attempted to simplify this management. Their objectives were to be more easily applicable in the clinical routine for general practitioners, and to provide more understandable messages for the patients [3,13]. The task force included gout experts from French referral centres with a dedicated recruitment of patients suffering from crystal-related arthropathies. The task force of the French 2020 recommendations for the management of gout decided to broaden the indication of ULT initiation to all patients with a diagnosis of gout and to set up an SU target of at least below 6.0 mg/dL and, if possible, below 5.0 mg/dL for all patients [3]. This had also been recommended by the British Society of Rheumatology guidelines [14]. However, to date no data have come forward on the feasibility and efficiency of applying these new recommendations in real-life care settings or their impact on patients’ outcomes.

French gout management is overwhelmingly performed in primary care, and referral centres are mostly meant to provide help in managing patients with difficult-to-treat (DTT) gout [15]. Difficult-to-treat gout encompasses diseases requiring specific ULT, such as uricosurics or uricases. It also includes gout flares that cannot be controlled with conventional therapy (colchicine, corticosteroids, and non-steroidal anti-inflammatory drugs) and that require interleukin-1 (IL-1) blockade, which entails initial in-hospital prescriptions in referral centres [16]. In France, canakinumab is the only IL-1-blocking drug with marketing authorisation for the treatment of gout flares. Its indication is restricted by the European (and French National) Medicines Agency (EMA) to non-responding patients or those intolerant to all conventional anti-inflammatory drugs, and those who are experiencing at least 3 flares per year, which is generally considered very restrictive [16]. To date, the profile of patients considered to have DTT gout in clinical practice has not been reported, which is a necessary step to fil the gap between clinical needs and regulatory authorisations.

The primary objective of the study was to describe first the profile: treatment over a two-year course and outcome of patients with gout managed by physicians from French referral centres, and second to what extent these referral centres applied the French recommendations eventually published in 2020. Secondary objectives included determining factors predictive of reaching SU levels below 6.0 mg/dL, and a description of anti-IL1 use.

## 2. Materials and Methods

### 2.1. Study Design and Population

This was a multicentre, retrospective, observational, and non-interventional study. One hundred patients attending a first visit for gout were consecutively included in each of the three French referral centres involved in the study (Lariboisière Hospital–Paris, Bichat Hospital–Paris, Saint Philibert hospital–Lille) if they fulfilled the following criteria: at least 18 years of age, diagnosis of gout established by a referral centre physician, and follow-up in the expert centre beginning between 1 January 2016 and 1 June 2019. Patients were not included if they were attending a one-time visit for advice (not for follow-up), or if there was doubt about the diagnosis. The first visit took place between 1 January 2016 and 1 June 2019 (index baseline visit). Data from baseline and the first two years of follow-up were recorded in computerised files by a referral centre physician as a part of the patient’s medical care, without a defined protocol. Then, they were retrospectively collected for the study between July 2019 and June 2021.

### 2.2. Baseline and Follow-Up Visits

At baseline, characteristics of patients were collected, including socio-demographic features, comorbidities, gout history, gout clinical characteristics, imaging data (ultrasound (US) and dual-energy computed tomography (DECT) scans to assess the monosodium-urate crystal (MSU) burden), and ongoing treatments that may affect SU levels. The context in which the baseline visit occurred (outpatient referral, or in-care hospitalization) was also recorded. At each visit, i.e., at baseline (M0), month 6 (M6), month 12 (M12), and month 24 (M24), were recorded: the number of flares in the past 6 months before inclusion/since the last visit (self-reported); ongoing treatments (flare prophylaxis, ULT); qualitative assessment of patient compliance, defined by taking the ULT at least 6 days a week, determined by the consulting physician on patient interview; and laboratory results including SU levels and estimated glomerular filtration rate (eGFR). Allopurinol doses were considered to be above restrictions based on kidney function if allopurinol doses were >300 mg/day when eGFR < 100 mL/min/1.73 m^2^, >200 mg/day when eGFR < 80 mL/min/1.73 m^2^, >100 mg/day when eGFR < 40 mL/min/1.73 m^2^, and >50 mg/day when eGFR < 20 mL/min/1.73 m^2^, as per French regulations for therapy maintenance (summary of the product characteristics used in France).

### 2.3. Statistical Analysis

Continuous variables were described as mean ± standard deviation, or median [Q1; Q3]. Categorical variables were described as numbers and frequencies. Some covariates were secondarily categorized (for details see Table 1) when relevant. Bivariate analysis was performed to compare (i) allopurinol versus febuxostat users, and (ii) patients lost to follow-up versus patients with at least 2 visits, using Student’s *t*-tests (or Mann–Whitney–Wilcoxon tests) and chi-square tests (or Fisher’s exact test), with a *p*-value significance set at 0.05. A logistic mixed model was used to study the predictive value of having an SU level below 6.0 mg/dL at each follow-up visit among patients undergoing more than one visit. First, bivariate logistic mixed models were run, including each variable of interest and the visit as fixed effects, and the patient as a random effect. ANOVAs were run for models with a categorical variable (>2 categories). Then, a multivariate logistic mixed model was obtained using an Akaike information criterion-based ascending variable-selection algorithm, starting with a model including only the visit as a fixed effect and the patient as a random effect. All variables with a *p*-value < 0.2 in the bivariate models were entered in the algorithm. A logistic mixed model predicting SU level below 5.0 mg/dL, including ULT (allopurinol or febuxostat) and the visit as fixed effects and the patient as a random effect, was also run. Statistical analyses were performed using R software (version 4.0.5). An arbitrary predefined number of 100 patients per centre was chosen to provide a representative overview of patients managed in the centre.

## 3. Results

### 3.1. Study Population

Three hundred patients were included and attended 488 follow-up visits (Figure 1). Global baseline characteristics of the patients are detailed in Table 1 (per-centre details in Appendix A). At baseline, the population was predominantly male (81%), with an average age of 62.2 ± 15.2 years, a mean duration of gout of 5.8 ± 8.6 years, and a median number of gout flares over the last 6 months of 1 [1; 2], with 94% of patients having experienced at least one flare. A total of 225 patients (75%) had gout complications (clinical/palpable subcutaneous or US tophi, eGFR < 60 mL/min/1.73 m^2^, kidney stones, gout arthropathy). The mean SU level at inclusion was 8.2 ± 2.4 mg/dL and only 81 patients (27%) had already been treated with ULTs. The first encounter was an outpatient visit for 190 patients (63.3%) and in-hospital care for the others. Overall, 178 patients (59.3%) were specifically referred to the centre for its expertise on gout, essentially from another department of the hospital (78.7%) or the on general practitioner’s initiative (15.2%). For those who were not referred, the context was a direct outpatient visit on the patients’ initiative (see Table 1).

### 3.2. ULT Management

At the end of the baseline visit, a ULT was prescribed in 254/300 patients (84.7%), mainly allopurinol (49.6%) and febuxostat (48%). For the others, it was not prescribed mainly due to a delayed introduction if the patient had a severe gout flare at the time of the medical visit, if there was initially a diagnosis doubt, or if dialysis or dietary rules were planned. The management proposed by the referral-centre physician at the baseline visit was consistent with the upcoming 2020 French recommendations in 94.9% of cases (among the others: 3.5% due to the ULT choice; 1.6% due to lack of a recent SU measurement). Regarding the follow-up, only 122/300 (40.7%) patients attended the M24 visit (Figure 1). Throughout this period, self-declared compliance with ULT, recorded by the rheumatologist, was 84.5%. SU levels < 5.0 mg/dL and <6.0 mg/dL were reached for 54.1% and 78.6% of patients at M24, respectively. There was no difference between allopurinol and febuxostat in the achievement of SU levels < 6.0 mg/dL at M24 (84.8% and 80.4% respectively, *p* = 0.78) (Figure 2 and Figure 3). However, febuxostat users (48.3%) had a 2.99 times greater chance (OR = 2.99, CI95% = [1.58; 5.68]) of reaching the target of SU < 5.0 mg/L than allopurinol users (51.7%) (Figure 2 and Figure 3). At M24, mean doses were 313 ± 105 mg/day for allopurinol (dose > 300 mg/d for 41%) and 84.4 ± 36.7 mg/day for febuxostat (dose > 80 mg/d for 20%). Among allopurinol users, the prescribed dosage exceeded eGFR restrictions in 24/39 (61.5%) at M24.

### 3.3. Flare Management

An initial assessment of patients’ MSU crystal burden at baseline was implemented in 180 (60%) of patients, using US (*n* = 90), DECT (*n* = 9), or both (*n* = 81). Flare prophylaxis was prescribed at M0 for 81.3% of patients (colchicine for 92.2% of patients) and at M24 for 23%. Between M12 and M24, 13.1% of patients still experienced at least one flare (Figure 4) and 9.1% still experienced pain from chronic synovitis (synovitis for more than 3 months). Regarding IL1-blockade prescription, overall, among patients experiencing at least one gout flare, 36 (12%) patients received it for the acute management of flares and 15 (5%) for flare prophylaxis. Anakinra was prescribed in all cases. Of these 50 (16.7%) patients, 18 met EMA requirements and 40 met the EULAR recommendations.

At baseline, the 10 patients who received IL1-blockade outside the EULAR guidelines were suffering from a disabling polyarticular flare (i.e., flare requiring hospital care), with or without a known gout (known gout diagnosis in 5/10 patients) and with or without chronic kidney disease (CKD) (eGFR < 60 mL/min in 6/10 patients). For these, anakinra was prescribed for 3–5 days, allowing a rapid recovery of autonomy and a return home. In 3/10 cases, anakinra was preferred over conventional anti-inflammatory therapy, with the objective of shortening the hospital stay and starting ULT straight away to improve compliance as opposed to a delayed introduction of ULT.

### 3.4. Predictive Factors of Reaching the Targeted Serum-Urate Levels

Predictive factors of having SU levels < 6.0 mg/dL were determined from the 209 patients with at least one follow-up visit (M6, and/or M12, and/or M24) and an available value for uricemia, representing a total of 405 follow-up visits. Among these visits, 271 (66.9%) displayed uricemia < 6.0 mg/dL, with 134 (33.1%) above the threshold. The bivariate analyses retained seven variables with *p* < 0.2 in the models (*p*-values from ANOVA = 0.15 for ethnicity and 0.43 for beverage consumption). Only one main predictor of reaching an SU level < 6.0 mg/dL was found in the multivariate analysis: allopurinol dose ≥ 300 mg/d or febuxostat dose ≥ 80 mg/d vs. none or ULT different from allopurinol and febuxostat or allopurinol < 300 mg/d or febuxostat < 80 mg/d (OR = 7.49, 95% CI = [2.17; 25.9]). Two other variables were closed to statistical significance: the presence of tophi on joint US (*p* = 0.053) and age at gout onset (per one year increase) (*p* = 0.052) (Table 2). Patient compliance with ULT was not included in the multivariate model because it was distorted, given that nearly all patients with SU levels < 6.0 mg/dL were considered compliant. This final model was obtained using 239 observations from 123 patients (*n* = 101 at M6, *n* = 79 at M12, *n* = 59 at M24). Of note, the rate of achievement of SU level < 6.0 mg/dL at M24 was similar among patients from all three referral centres throughout the follow-up.

### 3.5. Factors Predictive of Being Lost to Follow-Up

Patients lost to follow-up were divided into a first category, lost immediately after the baseline visit; and a second category, lost later in the follow-up due to several reasons: asymptomatic patients undergoing prophylactic treatment, patients with usual compliance difficulties, intercurrent health events, rheumatologist decision to stop the expert centre follow-up, and patients with cognitive disorders (see Appendix A).

A subgroup analysis was performed to compare the group of patients lost to follow-up just after baseline despite scheduled follow-up visits (*n* = 66, 22%) to the group of patients who attended at least one follow-up visit (*n* = 234, 78%). Two factors differed significantly between those groups: a first encounter as outpatient (versus in-hospital care) was significantly higher among the group of patients who attended at least one follow-up visit (67.5% vs. 48.5%, *p* < 0.01), as was disease duration (3 [0.5; 8] years vs. 1 [0; 5], *p* < 0.05), compared to the group of patients lost to follow-up (see Appendix A).

## 4. Discussion

The profile of patients managed in French gout-referral centres showed a few differences compare to the profile of those managed in primary care. In comparison to the 1003 patients of the French National Survey GOSPEL, who were managed by primary-care practitioners and rheumatologists in private practices, our patients displayed a similar prevalence of high blood pressure, dyslipidaemia, obesity, CKD stages 3–5, and gout duration, and diabetes mellitus was more prevalent (28.1% versus 15.0%) [1]. Our patients were also similar to the 3079 patients included in the CACTUS French and Greek primary-care cohort, except for a very low prevalence of CKD in the French patients of the cohort (9.5%) [18]. This is in line with the fact that in our study most patients were managed by referral-centre physicians either “by chance” because they had been hospitalised in the centre, or were referred for management that: only” required optimisation using conventional drugs, but not for truly difficult-to-treat gouts per se.

In our study, among those not lost to follow-up, gout management by physicians applying a clinical routine consistent with current recommendations enabled the vast majority of patients to reach SU level target < 6.0 mg/dL (78.6%), with increasing clinical improvement (i.e., flare decrease) over two years of follow-up, even in difficult-to-treat cases. To compare to the literature, the trial led by Doherty et al. involving a treat-to-target strategy implemented by nurses was pivotal in demonstrating that appropriate titration of ULT doses until reaching the SU target level led to an almost perfect rate of success with a dramatic improvement in clinical outcomes [8]. Indeed, at the end of the two-year follow-up, 95% of patients managed by trained nurses had SU levels < 6.0 mg/dL (with 88% reaching < 5.0 mg/dL) compared with 30% and 17%, respectively, in patients managed in a primary-care setting [8]. The rate of success was lower in our study than in the Doherty trial, but with a far lower number of visits for each patient inherent to the setting of referral centres, demonstrating that efficient management of gout can be performed by physicians as well. Therefore, the correct application, as well as the adherence to the 2020 French recommendations over two years, leads to a large majority of successful treatments of gout [3,13]. However, current results in primary care in France, similar to those observed in the United Kingdom, show only 34.5% of allopurinol-treated patients having SU levels < 6.0 mg/L [1]. The same results were found worldwide, as shown by a survey of approximately 850 private physicians in the United States, where management was in line with international recommendations for chronic gout in only 3% to 17% of patients [19]. Therefore, even if the Doherty et al. trial had finally demonstrated a concept, the question remained as to whether such results could be obtained by physicians working in their real-life clinical routine.

In our study, only one predictive factor of successful reaching SU levels < 6.0 mg/dL at a given visit in French referral centres was found: the need for a sufficient dosage of ULT as reminder of the need for proper titration. Although not included in the final model, compliance with ULT is logically associated with goal achievement: This predictive factor does not provide new knowledge but reaffirms the need for patient education (not recorded in our study because it is not systematically specified in the medical records). Age at diagnosis and high crystal load at inclusion, which were nearly significant in the multivariate model, suggest that it is never too late in life and in the disease course to achieve optimal management. Furthermore, the duration of gout at baseline was significantly longer in patients who attended multiple visits versus only one, probably due to greater motivation for getting rid of a long-lasting burden.

The comparison between allopurinol and febuxostat users, who remained on the same ULT drug during follow-up, supports the hypothesis that neither appears to be more effective than the other to determine the achievement of SU goals, as long as a treat-to-target strategy is applied, a finding also shown by the FAST study [20]. French recommendations provide some guidance on which ULT to initiate according to eGFR for reasons of cost and safety [3]. One must point out that this result implies the use of allopurinol doses above those restricted by kidney function in a substantial number of patients once the critical first six months of treatment for severe cutaneous reactions were safely through [21,22], as those restrictions are considered too stringent in daily practice. More febuxostat users than allopurinol users had final SU levels < 5.0 mg/dL, which reflects that referral-centre physicians were following EULAR guidelines [3] until a consensus was reached to advocate for a lower SU target (5.0 mg/dL) in the 2020 French recommendations, which presumably led to increased doses of allopurinol from 2020 onwards [3].

A very small proportion of patients in our cohort required specific ULT (namely, uricosurics), and the vast majority eventually only used xanthine-oxidase inhibitors, well known to primary care physicians. Therefore, the proper application of the recommendations in primary care would allow for the identification of truly difficult-to-treat gouts. The proportion of our patients considered to have DTT gout, with potential eligibility for IL-1 blockade (15.3% at baseline), was at least six-fold higher relative to those of the GOSPEL cohort, where they were encountered only exceptionally (10/1003 patients, according to the EMA requirements) [16]. In these French referral centres, IL-1-blockade use was completely off-label, as anakinra is not labelled for gout in France, even after canakinumab received marketing authorisation for gout flares in 2018. Furthermore, in our work anakinra was used outside the EMA restrictions for canakinumab use in more than half of the patients treated for flares and was used for flare prophylaxis for which canakinumab is not even labelled. This highlights the gap between real-life IL-1-blockade use in gout management for more than a decade and restricted labelling of IL-1 blocking agents by regulatory agencies [16,23]. In particular, this leaves open the question of a marketing authorisation for anakinra in gout at a time when new randomised controlled studies are attempting to prove its efficacy in treating gout flares [24,25]. In the rare case of patients with severe gout and a contraindication for colchicine, non-steroidal anti-inflammatory drugs, and steroids due to comorbidities (CKD 3–5, diabetes, etc.) or failure of treatment, it is not a matter of determining the superiority of anakinra but rather its efficiency and safety.

Finally, the use of advanced imaging techniques (US and DECT) was very common—although generally absent in primary care, it is explained by both its accessibility and the fact that these techniques are a specific field of research of these expert teams.

We acknowledge some limitations to our study. First, its retrospective nature, in particular the problem of inherent missing data that may have led to selection and reporting biases, although episodes of gout flares and SU levels are standard outcome measures.

Second, the content and time allocated to therapeutic information and education were not reported in patients’ medical files and could not be measured directly in our study, although their importance is emphasized by the latest recommendations [3,13] and some recent works such as those by Te Kampe et al. [26].

Third, the number of patients lost to follow-up was substantial and inherent to the referral-centre setting, designed more to provide expert advice than to achieve close follow-up management per se, which should more regularly be performed in primary care. Moreover, there was a probable bias due to the COVID-19 pandemic in the second part of the follow-up. However, in the referral centre setting, more time can be devoted to patient information and education, which is central to ensuring better compliance and overall self-appropriation of disease management [3,13] but less compatible with the pace of primary-care visits. This large number of dropouts reminds us that the interpretation of our results must be done with caution: The predictive factors for reaching the objective are valid for patients who are compliant with their gout monitoring.

Lastly, the fact that this study was only conducted in French referral centres may limit the generalizability of the results at an international level.

## 5. Conclusions

This study shows that that the characteristics of patients with gout, managed in French tertiary-care centres with expertise in gout management, are overall very similar to those seen in primary care, and that only a minority of them have DTT gout. All patients received optimal care simply because these centres apply current recommendations, also readily applicable in the primary-care setting. This shows that almost all patients with gout respond very well to simple and optimised management. This means that it is time to put a halt to the clinical and therapeutic inertia that, worldwide, flaws patient management for a rheumatic disorder that probably has the highest potential for achieving remission [5].

## Figures and Tables

**Figure 1 jcm-11-05742-f001:**
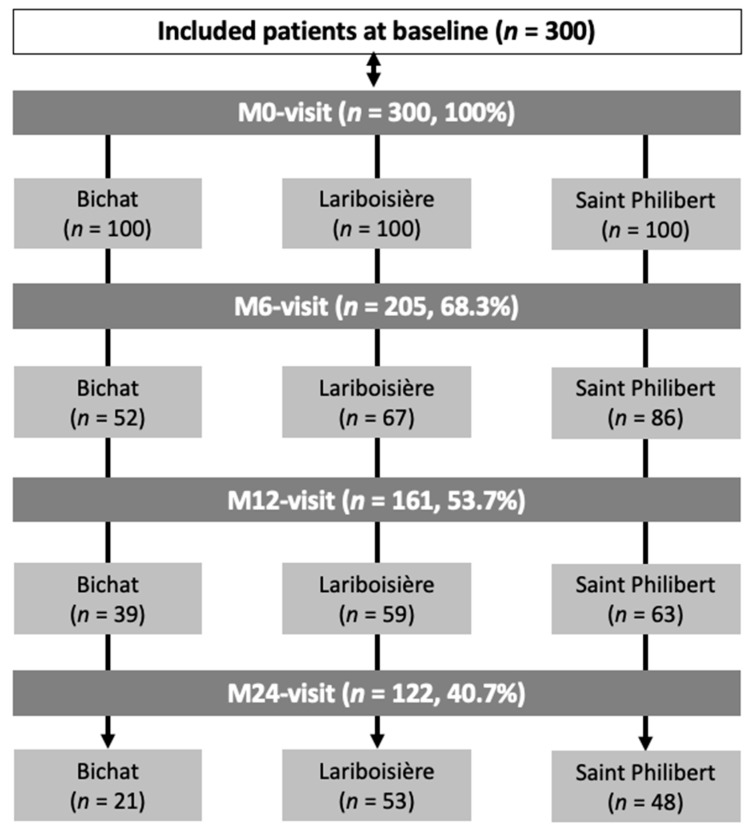
Study flow chart. M: month.

**Figure 2 jcm-11-05742-f002:**
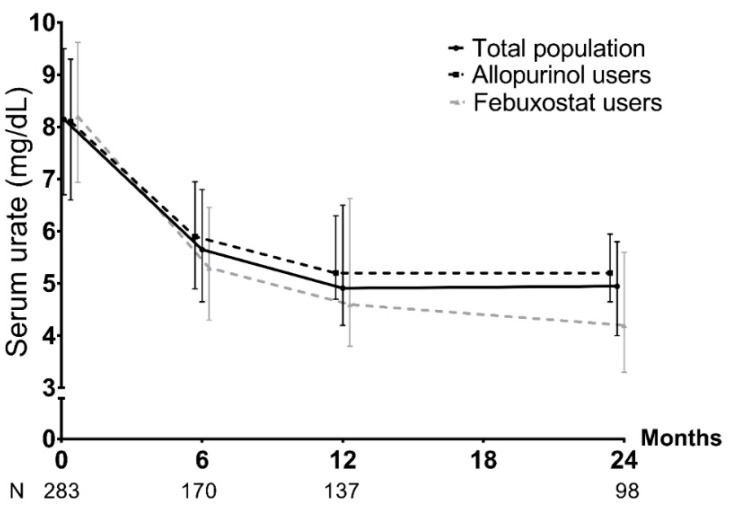
Evolution of median serum-urate level and interquartile range [Q1–Q3] during the follow-up. N = total number of patients with an available serum-urate measurement at the visit.

**Figure 3 jcm-11-05742-f003:**
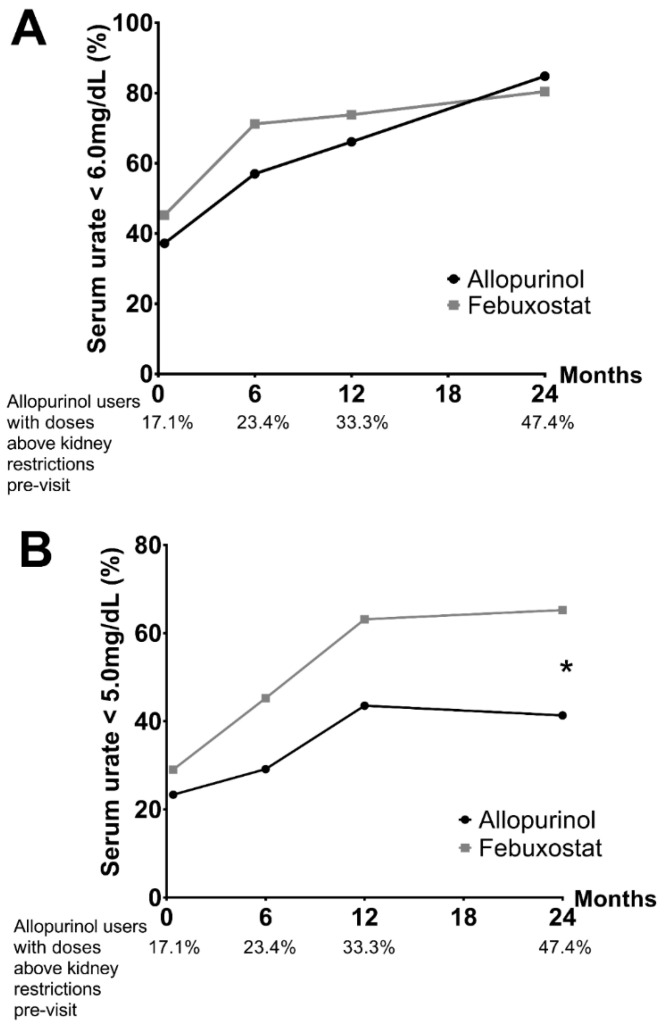
Percentage of patients treated with urate-lowering therapy reaching targeted serum-urate levels. Percentage of patients treated with febuxostat and allopurinol reaching serum-urate levels below (**A**) 6.0 and (**B**) 5.0 mg/dL, and percentage of allopurinol users treated with doses above restrictions according to estimated glomerular-filtration rates before the visit. * *p* < 0.01 in the mixed model.

**Figure 4 jcm-11-05742-f004:**
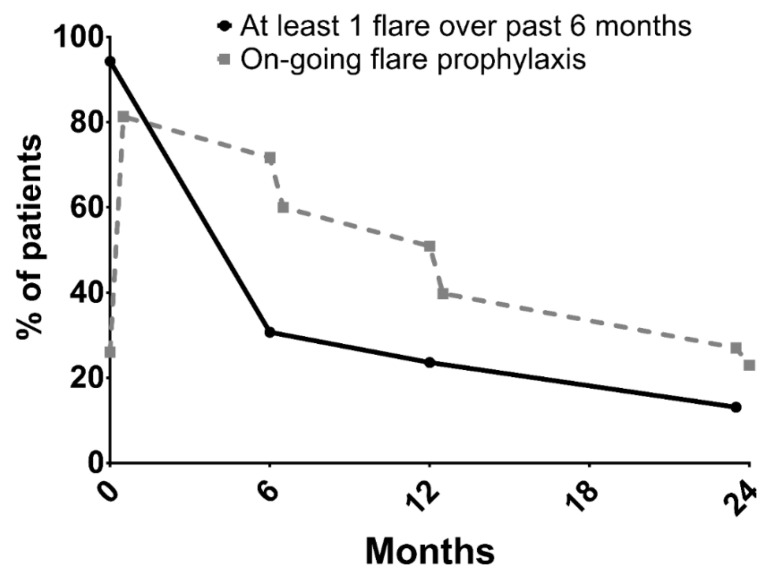
Evolution of gout-flare occurrence and flare prophylaxis before and after each follow-up visit.

**Table 1 jcm-11-05742-t001:** Patients’ baseline characteristics.

Baseline Characteristics		Missing Values (*n*)
Centre		0
Saint Philibert	100 (33.3%)	
Bichat	100 (33.3%)	
Lariboisière	100 (33.3%)	
Male	243 (81%)	0
Age at inclusion (years)	62.2 ± 15.2	0
Body-mass index (kg/m^2^)	27.9 ± 4.9	53
Ethnicity		9
Caucasian	163 (56%)	
North African	49 (16.8%)	
Sub Saharan African	48 (16.5%)	
Others	31 (10.7%)	
Socio-economic status		38
Managers	14 (5.3%)	
Academic professions	39 (14.9%)	
Workers, farmers	35 (13.4%)	
Unemployed	13 (5%)	
Home keeper, retired, or disabled	161 (61.5%)	
Beverage intake		1
No beverage consumption	194 (64.9%)	
Current alcoholic intoxication ^a^	55 (18.4%)	
Former alcoholic intoxication	26 (8.7%)	
Current excessive sweetened-beverage intake ^b^	7 (2.3%)	
Former excessive sweetened-beverage intake	3 (1%)	
Current alcoholic intoxication and excessive sweetened- beverage intake	13 (4.3%)	
Former alcoholic intoxication and excessive sweetened- beverage intake	1 (0.3%)	
Smoking status		1
No	190 (63.5%)	
Current smoker	41 (13.7%)	
Former smoker	68 (22.7%)	
Diet high in purine and/or fructose	116 (47%)	53
Practice of regular physical activity	31 (12.6%)	53
Rheumatologic and non-rheumatologic comorbidities		
Osteoarthritis	72 (24%)	1
Diabetes mellitus	84 (28.1%)	1
High blood pressure	181 (60.5%)	1
History of major cardiovascular-event history (stroke, myocardial infarction, lower-limb arteriopathy)	67 (22.3%)	1
Congestive heart failure	38 (12.7%)	1
Dyslipidaemia	100 (33.4%)	1
Liver disease	11 (3.7%)	1
Obesity (body-mass index > 30 kg/m^2^)	74 (30%)	53
Family history of ^c^		0
Gout (first or second degree)	55 (18.3%)	
Renal colic (first or second degree)	5 (1.7%)	
Hyperuricaemia (first or second degree)	2 (0.7%)	
Background treatments for comorbidities		
≥2 hyper-uricaemic treatments ^d^	126 (42.1%)	1
≥2 hypo-uricaemic treatments ^e^	78 (26.1%)	1
Kidney-failure treatment: dialysis, transplant	4 (1.3%)	1
Gout duration (years)	5.8 ± 8.6	14
Number of flares in the 6 months before baseline	1 [1; 2]	18
At least one gout complication at baseline	229 (76.3%)	0
If gout complication, type ^c^		0
Ultrasound or subcutaneous tophi	144 (62.9%)	
Kidney stones on imaging and/or renal colic	35 (15.3%)	
Chronic kidney disease (CKD 3 and above)	128 (55.9%)	
Gouty arthropathy	58 (25.3%)	
On-going urate-lowering therapy (ULT)	81 (27%)	0
If on-going ULT, name		0
Allopurinol	45 (55.6%)	
Febuxostat	34 (42%)	
Benzobromarone, Probenecid	2 (2.4%)	
Lesinurad	0 (0%)	
Rasburicase, Pegloticase	0 (0%)	
Bitherapy	0 (0%)	
Serum-urate level (mg/dL)	8.2 ± 2.4	17
First visit context Outpatient referral In-care hospitalisation	190 (63.3%) 110 (36.7%)	0
Patient specifically referred to expert centre	178 (59.3%)	0
If specifically referred, reason		
From primary care for treatment initiation	9 (5.1%)	
From another hospital department	140 (78.7%)	
Non-control at submaximal ULT dose in primary care	8 (4.5%)	
Non-control at a maximal dose of ULT in primary care	5 (2.8%)	
Non-control with non-referred management	1 (0.6%)	
For initial hospital prescription	1 (0.6%)	
Already follow in the expert centre for another reason	9 (5.1%)	
From primary care, for diagnosis re-evaluation	2 (1.1%)	
From primary care, for re-evaluation because of tolerance difficulties to standard ULT	1 (0.6%)	
For personal convenience	2 (1.1%)	
Mean ± SD, median [Q1; Q3], *n* (%).		

^a^ More than 21 units of alcohol per week in men, and more than 14 per week in women (WHO recommendations); ^b^ at least one serving per day [17]; ^c^ multiple choice; ^d^ beta-blocker, diuretics, aspirin; ^e^ losartan, calcium-channel blocker, atorvastatin, fenofibrate, ezetimibe.

**Table 2 jcm-11-05742-t002:** Predictive factors associated with serum-urate levels < 6.0 mg/dL at a visit in a referral centre.

	Univariate OR (95% CI)	Multivariate OR (95% CI)
Gout duration (years)	0.996 (0.95–1.04)	-
No ULT at baseline	1.38 (0.69–2.77)	-
Alcohol/sweetened-beverage consumption - No consumption - Former - Current	- 0.69 (0.21–2.29) 1.12 (0.52–2.39)	- - -
Ethnicity - Caucasian - North African - Sub Saharan African - Other	- 0.81 (0.31–2.12) 0.38 (0.12–1.15) ^a^ 0.36 (0.11–1.17)	- - - -
Body-mass index at baseline (kg/m^2^)	0.995 (0.93–1.07)	-
High blood pressure	1.47 (0.74–2.95)	-
Major cardiovascular event	0.99 (0.46–2.12)	-
Diabetes mellitus	1.24 (0.58–2.64)	-
Congestive heart failure	0.77 (0.26–2.32)	-
Dyslipidaemia	1.13 (0.55–2.31)	-
Hepatopathy	0.39 (0.07–2.05)	-
Number of gout complications at baseline	1.09 (0.77–1.55)	-
Ultrasound/subcutaneous tophi	2.63 (1.20–5.74) ^b^	3.00 (0.99–9.08)
Double contour sign identified with joint ultrasound	0.68 (0.32–1.48)	-
DECT MSU deposits at baseline	0.73 (0.22–2.43)	-
Age at gout onset (per year)	1.03 (1.01–1.06) ^b^	1.04 (1.00–1.08)
Age at baseline (per year)	1.03 (1.01–1.06) ^b^	-
Good compliance to ULT prior to visit:	35.36 (5.46–229.08) ^d^	-
ULT: - No ULT or ULT different from ALLO/FBX or ALLO < 300 mg/d or FBX < 80 mg/d - ALLO ≥ 300 mg/d or FBX ≥ 80 mg/d	- 9.74 (3.87–24.53) ^d^	- 7.49 (2.17–25.9) ^c^
Hyperuricaemic therapy added to the patient’s background treatment since last visit	0.78 (0.24–2.58)	-
Hyperuricaemic therapy removed from the patient’s background treatment since last visit	0.29 (0.07–1.10) ^a^	0.18 (0.027–1.22)
ULT switch at previous visit	0.83 (0.17–4.02)	-

*p*-value significance set at 0.05: ^a^*p* < 0.2, ^b^
*p* < 0.05, ^c^
*p* < 0.01, ^d^
*p* < 0.001; OR: odds ratio; CI: confidence interval; ALLO: allopurinol; FBX: febuxostat; ULT: urate-lowering therapy; DECT: dual-energy computed tomography; MSU: monosodium urate.

## Data Availability

The datasets analysed during the current study are available from the corresponding author on reasonable request.

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
