# Peer review of "Simple Application and Adherence to Gout Guidelines Enables Disease Control: An Observational Study in French Referral Centres"

_jcm, 2022, doi:10.3390/jcm11195742_

Round 1

Reviewer 1 Report

I commend Jauffret and colleagues for undertaking this important study: “Simple application of gout guidelines enables disease control: an observational study in French referral centres”. Please find my comments below mainly about the authors' data collection method. 

Abstract: Minor suggestion. Please add “study” to the sentence which states the objectives.  Currently it is not clear whether the objectives are for the study or SFR.

Methods: The study period needs to be stated clearly. Could you also detail the data collection method including who collected the data etc? What do you mean by “Data collected at baseline and follow-up were recorded in computerised medical files…”? Was the study part of patients’ medical care?

What was the role of the patient’s physician in this study?

I gather from the Informed Consent Statement that this was a retrospective study. However, it sounds more of  a primary study comprising of patient interviews e.g., the authors collected self-reported number of flares in the past 6 months and patient ULT compliance determined by the consulting physician on patient’s interview. Can you please clarify whether and how patient interviews were conducted?

Author Response

Point 1: “Abstract: Minor suggestion. Please add “study” to the sentence which states the objectives.  Currently it is not clear whether the objectives are for the study or SFR”.

Response 1: The modification has been done page 1 line 27. 

Point 2: “Methods: The study period needs to be stated clearly. Could you also detail the data collection method including who collected the data etc? What do you mean by “Data collected at baseline and follow-up were recorded in computerised medical files…”? Was the study part of patients’ medical care? What was the role of the patient’s physician in this study?

I gather from the Informed Consent Statement that this was a retrospective study. However, it sounds more of  a primary study comprising of patient interviews e.g., the authors collected self-reported number of flares in the past 6 months and patient ULT compliance determined by the consulting physician on patient’s interview. Can you please clarify whether and how patient interviews were conducted?”

Response 2: We thank the reviewer for the suggestions. Patients were included in the study if their first visit took place between January 1, 2016 and June 1, 2019 (index baseline visit).

Data were collected by the treating physician during the baseline and follow-up visits  in computerized files as per usual  patient's medical care. Data from baseline and follow-up visits of the first 24 months of management were then retrospectively collected for the study by CJ, between July, 2019 and June, 2021.

The medical interviews were conducted as standard rheumatology visits, without a defined protocol, and without any ulterior motive for inclusion in the study, which was started later.

Modifications have been done page 3 line 101 to 105. 

Reviewer 2 Report

When tophaceous gout is 62.9%, and CKD is 55.9%, the proportion is higher than that reported in existing population-based studies, so it is difficult to agree with the author's statement that profiles are not significantly different from primary care.

However, even in difficult-to-treat gout, the result that disease control was achieved in 78.6% of cases simply by consistently following the guidelines rather than special techniques would be an excellent message to encourage treating physicians.

One issue I would like to raise is whether the title or conclusion of this manuscript is in line with the results of this study. 

When 300 patients started treatment and only 122 x 78.6% of patients have confirmed to achieve the treatment goal, it is hard to say the disease can be well controlled if only the doctor properly applies the French guidelines. Instead, the results of this study are more suitable for 'the adherence of gout guidelines over 12 months (or two years) enables fair disease control in difficult to treat gout'.

From this point of view, because this study's result ultimately speaks to the importance of adherence, it is necessary to analyze further and discuss the 60% of patients lost to follow-up to increase this study's clinical impact. I believe that the characteristics of patients with follow-up loss are worth organizing in a separate table.
